# Hypopituitarism, Diabetes Insipidus, and Syndrome of Inappropriate Antidiuretic Hormone Secretion after Pituitary Macroadenoma Surgery with Indocyanine Green Dye

**DOI:** 10.3390/diagnostics14171863

**Published:** 2024-08-26

**Authors:** Tomislav Felbabić, Tomaž Velnar, Tomaž Kocjan

**Affiliations:** 1Department of Neurosurgery, University Medical Centre Ljubljana, 1000 Ljubljana, Slovenia; tvelnar@hotmail.com; 2Faculty of Medicine, University of Ljubljana, 1000 Ljubljana, Slovenia; 3Department of Endocrinology, Diabetes and Metabolic Diseases, University Medical Centre Ljubljana, 1000 Ljubljana, Slovenia

**Keywords:** pituitary, adenohypophysis, neurohypophysis, non-functioning macroadenomas, ICG dye, endoscopic endonasal surgery, diabetes insipidus, pituitary–peripheral axis

## Abstract

(1) Background: Pituitary adenomas are benign tumors comprising about 18% of all intracranial tumors, and they often require surgical intervention. Differentiating pituitary tissue from adenoma during surgery is crucial to minimize complications. We hypothesized that using ICG dye would reduce the hormonal complication rates. (2) Methods: A prospective randomized study (February 2019–October 2023) included 34 patients with non-functional macroadenomas of the pituitary gland randomly assigned to receive intraoperative ICG or be in the control group. All underwent endoscopic endonasal transsphenoidal surgery. Pituitary function was assessed preoperatively, immediately postoperatively, and 3–6 months postoperatively. Adenohypophysis function was evaluated with hormonal tests (Cosyntropin stimulation test, TSH, fT3, fT4, prolactin, IGF-1, FSH, LH, and testosterone in men) and neurohypophysis function with fluid balance, plasma and urine osmolality, and serum and urinary sodium. (3) Results: Of the 34 patients (23 men, 11 women; average age 60.9 years), 5.9% in the ICG group developed diabetes insipidus postoperatively, compared to 23.5% in the control group. Adenohypophysis function worsened in 52.9% of the ICG group and in 35.3% of the control group. (4) Conclusions: Our study did not confirm the benefits of using ICG in these surgeries. Further research with a larger sample is needed.

## 1. Introduction

Pituitary adenomas are benign tumors of the anterior lobe of the pituitary gland. They account for approximately 18% of all intracranial neoplasms and are the most common tumors of the sellar region [1]. The prevalence of the disease in the general population is estimated at 16.7% [2]. They are the third most common intracranial neoplasm requiring surgery after gliomas and meningiomas [1]. Most pituitary adenomas are incidental findings and do not require intervention other than follow-up. Prolactinomas are most common (30–60%), followed by nonfunctioning adenomas (14–55%), growth-hormone-secreting adenomas (8–15%), and corticotropin-secreting adenomas (2–6%). Thyrotropin- and gonadotropin-secreting adenomas are extremely rare (<1%) [3]. Most of them occur sporadically (95%). A small minority are familial or part of a specific syndrome, such as multiple endocrine neoplasia types 1 and 4, McCune–Albright syndrome, Carney complex, and X-linked acrogigantism [4].

Pituitary adenomas are classified according to their size and hormonal overproduction. Those that are less than 1 cm in size are called microadenomas, those that are 1 cm or more in size are called macroadenomas, and those that are more than 4 cm in size are called giant [5]. Those pituitary adenomas that do not secrete hormones or whose hormone secretion does not cause clinical symptoms are called nonfunctioning adenomas. Hormonally active adenomas are referred to as functioning adenomas [6].

More sophisticated classifications have been developed to describe tumor invasion into the sella-adjacent structures. The Hardy classification describes the invasion of the sella floor [7]. A modified version by Willson, known as Hardy–Willson classification, describes the suprasellar invasion, but it lacks utility for the purposes of surgical cure and complication estimation [8]. Knosp’s classification describes invasion of the parasellar space [9].

Patients with pituitary adenoma should be evaluated by a multidisciplinary team including endocrinologists, ophthalmologists, and neurosurgeons. Surgical therapy is indicated in cases where adenomas cause neurologic deficits due to the mass effect of the lesion, show growth on follow-up MRI, or cause hormonal dysfunction [10]. An exception is prolactionoma, where the first stage of treatment is conservative with dopamine agonists [11]. Historically, pituitary adenomas were removed through a transcranial approach followed by a microscopic transsphenoidal approach. In recent years, the endoscopic endonasal transsphenoidal approach has become the gold standard for pituitary adenoma surgery [12].

It is very important to distinguish normal pituitary tissue from adenoma. In this way, maximal resection of the tumor can be achieved safely and with little risk of postoperative complications due to pituitary damage. Histologic and imaging studies have shown that normal pituitary tissue and pituitary adenoma have different vascular densities [13]. Adenomas have significantly lower capillary density compared with normal pituitary tissue. Advances in intraoperative tumor imaging have demonstrated that the use of indocyanine green (ICG) is of great benefit in endoscopic endonasal transsphenoidal pituitary adenoma surgery [14]. After intravenous application, the dye binds to serum albumin and causes fluorescence of vascular structures. The maximum emission wavelength of ICG in plasma is 820 nm [15]. Due to the dense vasculature of the pituitary gland and the small vasculature of the adenoma, ICG is a promising pituitary marker [16].

In this study, we hypothesized that because of the easier intraoperative differentiation between pituitary adenoma and the normal gland through interoperative ICG, the incidence of postoperative hypopituitarism and fluid balance disorders should be lower. If this is true, ICG will prove to be a valuable adjunct to endoscopic endonasal pituitary adenoma surgery as it will provide better outcomes for patients.

## 2. Materials and Methods

Patients

This prospective randomized study was conducted at the Departments of Neurosurgery and of Endocrinology, Diabetes, and Metabolic Diseases, University Medical Centre (UMC) hospital in Ljubljana, Slovenia. It was approved by the Slovenian Medical Ethics committee (No. 0120-56/2019/6). In total, 34 patients were included in the study during the period between February 2019 and October 2023. Patients were randomly placed in two groups: a group that received ICG intraoperatively and a control group. All patients signed written informed consent forms for inclusion in this study. An endoscopic endonasal transsphenoidal approach for pituitary adenoma removal was performed on all patients.

Indocyanine green material

The ICG compound was acquired from Serb pharmaceuticals, Paris, France. First, 25 mg of the compound was dissolved in 10 mL of sterile water. After exposure of the intrasellar space, 5 mL of solution (12.5 mg of ICG) was injected intravenously as a bolus during surgery and after macroscopic removal of the macroadenoma. 

Optics

At our institution, we use the Karl Storz rigid endoscope for endoscopic transsphenoidal surgeries. For the purpose of this study, an endoscope with ICG light detection was used. The light source can be switched between white and near-infrared light intraoperatively using a foot switch.

Hormonal testing

At the Department of Endocrinology, Diabetes and Metabolic Diseases of the UMC Ljubljana, pituitary function was examined before surgery, immediately after surgery, and 3–6 months after surgery. The function of the pituitary–peripheral axis was determined through routine hormone tests: Cosyntropin stimulation test with cortisol basal and after 30 min, thyrotropin (TSH), free triiodothyronine (fT3), free thyroxine (fT4), prolactin (PRL), insulin-like growth factor-1 (IGF-1), follicle-stimulating hormone (FSH), luteinizing hormone (LH), and, in men, testosterone. We measured the fluid balance, the osmolality of the first morning urine, plasma osmolality, and the serum sodium with the urinary sodium in order to check water metabolism and the function of the neurohypophysis.

Hypopituitarism was defined as a condition in which one or more aspects of the pituitary–peripheral axis was affected and the patient required hormone replacement therapy. The assessment of hypopituitarism immediately after surgery is approximate, as all patients receive prophylactic doses of hydrocortisone (10 mg at 8.00, 5 mg at 13.00, and 5 mg at 17.00). Improvement was defined as when the therapy was reduced 3–6 months after surgery compared with the preoperative or postoperative state. If the therapy 3–6 months postoperatively was equal to the preoperative or postoperative therapy, we defined the condition as being as stable as before surgery or after surgery. However, if the patient required higher doses of therapy 3–6 months postoperatively compared with the preoperative or postoperative state, we classified it as deterioration. Diabetes insipidus was defined as a condition where the patient required sublingual desmopressin therapy due to polyuria. In contrast, syndrome of inappropriate antidiuretic hormone secretion was defined as a condition in which the patient required fluid restriction due to low serum sodium levels in combination with other established diagnostic criteria.

Radiologic evaluation

Contrast-enhanced MRI imaging of the head was performed preoperatively and 3–6 months postoperatively in all patients according to the sella region pathology protocol. Tumor dimensions were measured in the transverse, sagittal, and coronal planes. Tumor volume was calculated as half of the product of all three dimensions. Knosp classification was determined using the coronal sequences, while the Hardy classification was determined using the sagittal sequences.

Statistical analysis of the data

We analyzed the data using the statistical software SPSS 25 (IBM Corp., Armonk, NY, USA). We considered *p*-values of less than 0.05 to be statistically significant.

Demographic and clinical characteristics of the patients, which were normally distributed, are presented with the mean and the standard deviation, while asymmetrically distributed variables are presented with the median and the first and third quartiles. The values of the descriptive variables are presented as relative and absolute frequencies.

To compare the proportions between the patient groups with and without ICG dye, we used the chi-square test or Fisher’s exact test. The comparison of numerical variables between the study groups was performed using the *t*-test for independent samples or the non-parametric Mann–Whitney test. To compare the frequencies, we used the extended Fisher’s exact test according to Freeman–Halton. To compare the total number of axes affected to the time of the control measurement, we used the dependent samples *t*-test and the Wilcoxon signed ranks test.

## 3. Results

In total, 34 patients were included in this study, 23 males (67.7%) and 11 females (23.3%), aged between 33 and 81 years, with an average of 60.9 years. Demographic and clinical data are presented in Table 1. There were no significant differences between the groups based on the patients’ age, sex, or Knosp or Hardy classifications of the tumor. The sizes of tumors showed statistically significant differences.

We assessed the function of the neurohypophysis by evaluating fluid balance, the osmolality of morning urine, the plasma osmolality, and the serum sodium with the urinary sodium. Table 2 shows a comparison of the two quantities measured between the two groups of patients and between the controls.

The presence of hypopituitarism was assessed at all three control measurements, i.e., before surgery, immediately after surgery and 3–6 months after surgery. The incidence of hypopituitarism, shown separately according to the patient group, is shown in Table 3. If hypopituitarism was compared between all three measurements, the output was categorised in five ways ^2^. However, it is much less complicated if we only compare the preoperative state with the 3–6-month postoperative state. If we only compared those two states, the result was categorized in three ways ^3^.

We compared the number of affected pituitary–peripheral axes before surgery and 3–6 months after surgery. A statistically significant difference in the number of affected axes between the two groups was confirmed in the control 3–6 months after surgery. No significant difference in the number of affected axes between the groups was confirmed in the preoperative examination (Table 4).

We also compared the mean number of affected axes between the two groups before surgery and 3–6 months after surgery. The number of affected axes was statistically significantly different between the two control groups: 3–6 months after surgery, the mean number of affected axes was statistically significantly higher (1.82) than before surgery (1.24). We did not confirm a difference in the mean number of axes affected between the two patient groups (Table 5).

We were also interested in how often each type of pituitary–peripheral axis is affected before surgery and 3–6 months after surgery. The results are presented in Table 6.

Below are graphs comparing pituitary–peripheral axis involvement before and 3–6 months after surgery for all patients (Figure 1), the ICG group (Figure 2), and the control group (Figure 3) according to Table 6.

We were interested in the impact of demographic and clinical characteristics of patients on the outcome of pituitary adenoma surgery. The outcome was defined as the presence of the following conditions: hypopituitarism 3–6 months after surgery (yes vs. no) and diabetes insipidus 3–6 months after surgery (yes vs. no). The results are shown in Table 7.

## 4. Discussion

The use of ICG dye in pituitary macroadenoma surgery is a relatively new technique. Some authors have already described the potential benefits of using this method [16,17,18,19,20,21]. We present the first prospective randomized trial comparing the outcomes of endoscopic endonasal transsphenoidal surgery with and without the use of ICG. 

In our study, we did not find a better result of the operations with ICG dye compared to the conventional method. Litvac et al. have already shown in 2012 that ICG is a good potential marker for the pituitary gland [16]. This was confirmed in our study, as in all 17 patients who received ICG during surgery, the pituitary gland started to fluoresce after about half a minute, while the tumor tissue remained unstained. This is the so-called “early window”. Jeon et al. described the concept of a late or second window [22]. If the patient receives the dye 16–30 h before surgery, it is thought that the dye is absorbed into the adenoma due to the increased permeability of the tumor microcirculation, but this was not tested in our study. Despite the fact that the pituitary gland stains better with ICG and is therefore easier to distinguish from the tumor, we did not find any statistically significant differences in postoperative outcomes between the two groups in our study. The differentiation between the adenoma and the pituitary gland is already reliable under white light. The adenoma tends to be pale grey, softer, and easier to aspirate [23,24]. Therefore, the surgical technique was not significantly different between the two groups. However, in the few cases in the control group, we encountered a piece of tissue when removing the adenoma where we were not sure whether it was healthy pituitary tissue or a tumor. In such cases, we are in a dilemma: either we remove the suspicious tissue and risk hypopituitarism or we leave it and risk a remnant or recurrence. These are the cases where the potential benefit of ICG could be demonstrated. However, as there are few such cases, the sample size would need to be much larger to collect a sufficient number of doubtful cases.

In our study, we did not find a lower incidence of diabetes insipidus with the use of ICG dye. Diabetes insipidus, which may be transient or permanent, is one of the more common complications of endoscopic transsphenoidal surgery for pituitary macroadenomas. Zhan et al. published a large study of a total of 313 patients in whom postoperative complications were observed after endoscopic surgery for nonfunctioning adenomas [25]. They observed that diabetes insipidus occurred in 15.6% in the early postoperative period, while persistent diabetes insipidus was observed in 3.8%. In our study, we observed a total of seven (20.6%) cases of early postoperative diabetes insipidus, of which three (17.6%) were in the ICG group and four (23.5%) were in the control group. Permanent diabetes insipidus was present in one (5.9%) patient in the ICG group and in four (23.5%) patients in the control group. The proportion of diabetes insipidus in the ICG group is comparable to the results of Zhan’s study, but a higher proportion of patients with both transient and permanent insipidus was observed in the control group. Because Zhan et al. also included 53 (16.9%) patients with pituitary microadenomas, it is not unexpected that they obtained slightly better results. All of our patients had macroadenomas. The incidence of diabetes insipidus after endoscopic pituitary macroadenoma surgery was also studied by Kadir et al. [26]. They included 33 patients, which is almost identical to our sample size, so the results are more comparable. They described a 33.4% incidence of diabetes insipidus in the early postoperative period. Persistent diabetes insipidus was not described. In our study, as already stated above, 20.6% of patients had diabetes insipidus in the early postoperative period, of which 3/17 (17.6%) were from the ICG group and 4/17 (23.5%) were from the control group. Permanent diabetes insipidus was present in 1/17 (5.9%) patients from the ICG group and in 4/17 (23.5%) patients from the control group. Nevertheless, the statistical comparison of the samples showed no significant difference between the two groups, as shown in Table 2. We assume that a larger sample of patients could have shown a statistically significant difference. In their study of 271 patients with pituitary adenoma undergoing endoscopic surgery, Nayak et al. found that tumor size, suprasellar growth, and preoperative visual disturbances were positive predictors of the development of postoperative diabetes insipidus [27]. Our study partially confirmed this finding, as we found suprasellar tumor growth in four out of a total of five patients with persistent diabetes insipidus. Preoperative visual field loss was present in three of the five patients, but, in contrast, only one of our five patients with persistent diabetes insipidus had a tumor larger than the average of our sample.

In addition to diabetes insipidus, we also recorded the occurrence of SIADH. According to literature data, SIADH occurs in 4–23% of cases after endoscopic surgery of pituitary adenomas [28,29,30]. It is usually transient and resolves within a few weeks. In our study, we found three (8.8%) cases of SIADH, with two in the ICG group and one in the control group (Table 2). In all cases, it disappeared shortly after surgery. Our results regarding SIADH are therefore comparable with the literature. We could not find a statistically significant difference between the two groups, so we cannot claim that the intraoperative use of ICG leads to a better outcome regarding the occurrence of SIADH compared to the conventional method.

We also found no statistically significant difference between the two groups with regard to adenohypophysis function. In the literature, postoperative hypopituitarism has been reported in 1.4 to 19.8% [25]. In our study, the incidence of this complication was higher (Table 3). Preoperatively, the pituitary gland was functioning normally in only 14 of 34 patients (41.2%) and 3–6 months postoperatively in only 8/34 patients (23.5%). New-onset hypopituitarism was therefore noted in 6/14 patients (42.9%), 3 of whom were in the ICG group and 3 of whom were in the control group.

We were interested in the number of pituitary–peripheral axes affected (Table 4). Overall, 9/17 (52.9%) patients in the ICG group and 6/17 (35.3%) in the control group deteriorated after surgery. There is also a statistically significant difference in the number of affected axes between the two groups 3–6 months after surgery. However, upon closer analysis, we find that this statistically significant difference only means that the frequencies are differently distributed between the ICG and non-ICG groups, with the ICG group having a significantly higher proportion of cases of 0 and 3 pituitary–peripheral axis involvement, while the control group has a significantly higher proportion of 2 axis involvement. It cannot be argued that the final outcome of one group is better or worse compared to the other. When we compare the mean values of the affected axes, we see that there is no statistically significant difference between the groups (Table 6). A statistically significant difference occurs when comparing the mean number of affected axes of all patients before surgery with the mean number of affected axes after surgery. This means that our patients had a proven higher incidence of hypopituitarism 3–6 months after surgery regardless of the use of ICG. We also looked at the involvement of each pituitary–peripheral axis (Table 6). Mavromati et al. published a study in which they examined the surgical outcome of nonfunctioning pituitary macroadenomas according to the number of axes involved [31]. Patients were operated on using a combined microscopic–endoscopic approach. Of the 137 patients, 33% had no hypopituitarism preoperatively, 62.4% had hypogonadism, 41% had hypothyroidism, 30.8% had adrenal axis insufficiency, and 29.9% had growth hormone deficiency. Meanwhile, 3–6 months after surgery, 10% of patients had a new insufficiency of at least one of the pituitary–peripheral axes, 40.8% had hypogonadism, 29.3% had hypothyroidism, 27.4% had adrenal insufficiency, and 17% had growth hormone deficiency. In our study, we came to different results. The proportion of defects in the pituitary–peripheral axis before surgery is comparable, with 44.2% having hypogonadism, 44.1% with hypothyroidism, 29.4% with adrenal insufficiency, and 5.9% with growth hormone deficiency. However, 3–6 months after surgery, all of our axes showed deterioration in contrast to the study published by Mavromati et al. Postoperatively, 52.9% had hypogonadism, 55.9% had hypothyroidism, 52.9% had adrenal insufficiency, and 20.6% had growth hormone deficiency. There were no significant differences between our two groups. For each axis, approximately half of the patients were from the ICG group and the other half were from the control group.

We also investigated the influence of patient demographic and clinical characteristics on the outcomes of surgery for nonfunctioning pituitary macroadenomas, independent of the use of ICG. We were interested in hypopituitarism and diabetes insipidus as endpoints. The results are shown in Table 7. We were unable to demonstrate statistical significance in any of the parameters measured. However, borderline statistical significance (*p* < 0.1) was found in some cases. Regarding age, we cannot claim that it has an influence on the occurrence of hypopituitarism and diabetes insipidus. For gender and tumor size, we could not even detect borderline statistical significance. However, tumor size approached borderline statistical significance for postoperative hypopituitarism outcomes. For every one increase in the Knosp classification level, the probability of hypopituitarism 3–6 months after pituitary adenoma surgery increased 4.15-fold. Similarly, the probability of hypopituitarism 3–6 months after pituitary adenoma surgery increased 2.48-fold for every one increase in Hardy classification level. The Knosp and Hardy classifications showed no borderline significance for diabetes insipidus.

The main weakness of our study is the small number of patients included, which is partly due to the COVID-19 pandemic, when many patients with pituitary macroadenomas did not undergo surgery [32]. In addition, our results were affected by the initial inexperience of the surgical team in using the new equipment supporting ICG optics. We used a no-rinse endoscope, and it was often necessary to manually wipe the endoscopic camera during surgery, which disrupted the thread of the procedure. Also, when using ICG dye, the endoscope is slightly thicker than a conventional endoscope, so it provides less room for maneuvering, which can complicate the surgery. Another weakness of the study is the statistically significant higher volume of macroadenomas in the control group despite randomization in the group. We calculated the volume by measuring the largest dimensions of the tumor in the axial, sagittal, and coronal planes through magnetic resonance imaging according to the protocol for pituitary tumors, multiplied these measurements, and divided the result by half [33]. The mean volume in the ICG group was 5130 mm^3^, and in the control group it was 10473 mm^3^. When comparing the individual cases, we found that the largest case clearly stood out in terms of tumor size. The volume of the largest case was 34,965 mm^3^, while the average volume was 7802 mm^3^, which is more than four times smaller. When we excluded the largest case from the analysis, there was no longer a statistically significant difference in volume between the groups. Below are coronal images showing the largest case and a case that was closest in volume to the average value (Figure 4).

The advantage of our study is that all patients were operated on by the same neurosurgeon, so there could not have been different results between the patients due to the different experience of the surgeons. Furthermore, the surgical approach and technique were always the same. All of the diagnostic laboratory and radiological measurements of the study are standard and performed in all related institutions, so the reproducibility and comparability of the study with other studies of this kind are very simple and practically perfect. It is also worth emphasizing once again that, to our knowledge, we present the first prospective, randomized study comparing conventional and fluorescence-guided endoscopic transsphenoidal surgery for pituitary macroadenomas. We hope that it will be a trigger for further research in this field.

## 5. Conclusions

The results of our study did not confirm that the use of ICG dye in endoscopic endonasal transsphenoidal surgery of nonfunctioning pituitary macroadenomas reduces the incidence of hypopituitarism, diabetes insipidus, or SIADH. Therefore, none of our hypotheses can be confirmed. We suggest that the benefit of ICG dye in this type of surgery could be better evaluated in prospective, randomized, and preferably multicenter trials with a larger number of patients.

## Figures and Tables

**Figure 1 diagnostics-14-01863-f001:**
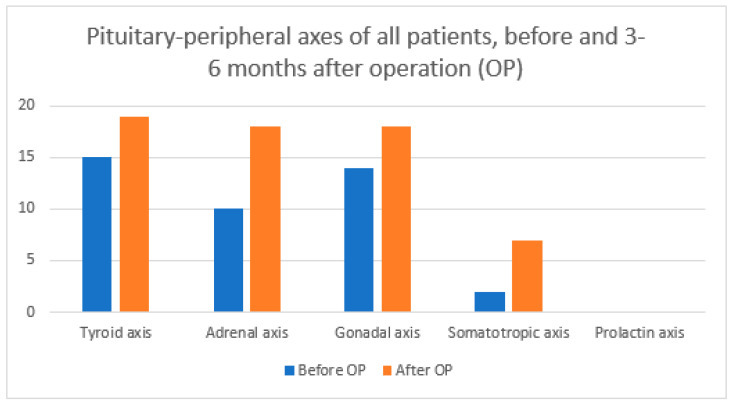
Pituitary–peripheral axis involvement before and 3–6 months after OP for all patients according to Table 6. OP = operation.

**Figure 2 diagnostics-14-01863-f002:**
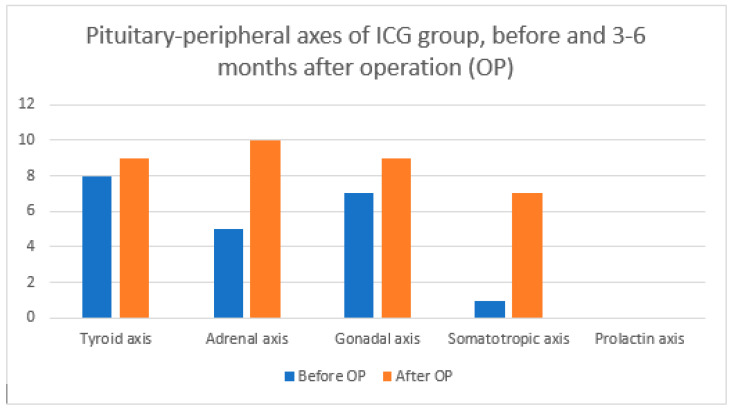
Pituitary–peripheral axis involvement before and 3–6 months after OP for ICG group according to Table 6. OP = operation.

**Figure 3 diagnostics-14-01863-f003:**
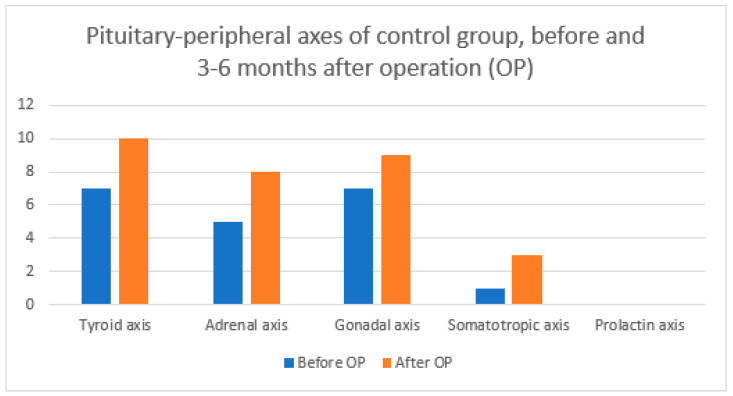
Pituitary–peripheral axis involvement before and 3–6 months after OP for control group according to Table 6. OP = operation.

**Figure 4 diagnostics-14-01863-f004:**
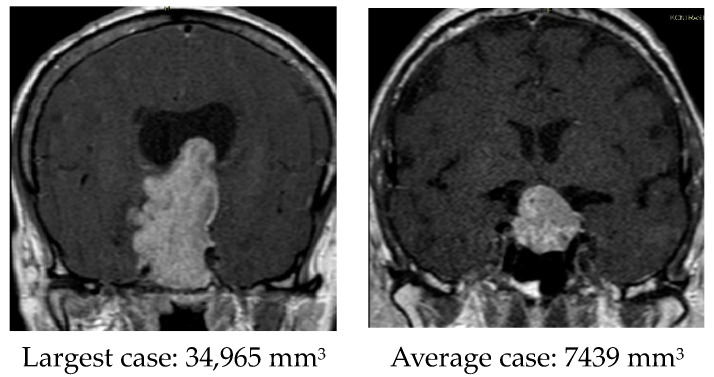
Coronal images showing the largest case (**left**) and average case (**right**).

**Table 1 diagnostics-14-01863-t001:** Demographic and clinical characteristics of patients with nonfunctioning pituitary adenoma, stratified by ICG application.

	All n = 34	ICG n = 17	No ICG n = 17	*p*-Value ^1^
Age, years	60.9 ± 11.2	62.7 ± 9.9	59.1 ± 12.4	0.360
Gender, n (%)				0.714
Male	23 (67.7)	11 (64.7)	12 (70.6)	
Female	11 (32.4)	6 (35.3)	5 (29.4)	
Tumor size, mm^3^	7802 (2577–11,261)	5130 (2344–6026)	10,473 (3753–16,273)	0.041
Knosp classification, n (%)				0.764
0	3 (8.8)	2 (11.8)	1 (5.9)	
1	18 (52.9)	8 (47.1)	10 (58.8)	
2	8 (23.5)	5 (29.4)	3 (17.6)	
3	5 (14.7)	2 (11.8)	3 (17.6)	
Hardy classification, n (%)				0.817
1	5 (14.7)	3 (17.6)	2 (11.8)	
2	9 (26.5)	4 (23.5)	5 (29.4)	
3	10 (29.4)	6 (35.3)	4 (23.5)	
4	10 (29.4)	4 (23.5)	6 (35.3)	

Numerical variables are presented as mean ± standard deviation or median (first–third quartile). ICG = indocyanine green dye. ^1^
*p*-value for comparison between patient groups with ICG and without ICG.

**Table 2 diagnostics-14-01863-t002:** Fluid balance before surgery, immediately after surgery, and 3–6 months after surgery.

	ICG Group			No ICG Group			*p* _1_	*p* _2_	*p* _3_
	Pre-OP	Post-OP	3–6 Months Post-OP	Pre-OP	Post-OP	3–6 Months Post-OP	ICG vs. No ICG	Control Measurement	Interaction
Fluid balance, n (%)							0.210	0.116	0.258
No complication	17/17 (100)	12/17 (70.6)	16/17 (94.1)	17/17 (100)	12/17 (70.6)	13/17 (76.5)			
Complication	0/17 (0)	5/17 (29.4)	1/17 (5.9)	0/17 (0)	5/17 (29.4)	4/17 (23.5)			
Fluid balance complication, n (%)							-	-	
Diabetes insipidus	-	3/5	1/1	-	4/5	4/4	0.148	0.399	0.399
SIADH	-	2/5	0/1	-	1/5	0/4	-	-	

ICG = indocyanine green dye, OP = operation, SIADH = syndrome of inappropriate antidiuretic hormone secretion, *p*_1,2,3_ = influence of ICG dye and control measurement time (only post-OP and 3–6 months post-OP) on fluid balance complication in a generalized linear model, *p*_1_, *p*_2_ = statistical significance of main effects, *p*_3_ = statistical significance of interaction between main effects.

**Table 3 diagnostics-14-01863-t003:** Incidence of hypopituitarism before adenoma surgery, immediately after surgery, and 3–6 months postoperatively after OP, separated according to ICG application.

	ICG n = 17	No ICG n = 17	*p*-Value ^1^
HP before OP, n (%)			0.3
No	8 (47.1)	6 (35.3)	
Yes	9 (52.9)	11 (64.7)	
HP after OP, n (%)			-
No	0 (0)	0 (0)	
Yes	17 (100)	17 (100)	
HP 3–6 months after OP ^2^, n (%)			0.822
Not detectable	5/17 (33.3)	3/17 (18.8)	
Improvement	2/17 (13.3)	2/17 (12.5)	
As after OP (HP already before OP)	1/17(6.7)	3/17 (12.5)	
After OP (HP occurs after OP)	1/17 (6.7)	3/17 (18.8)	
Deterioration	8/17 (40.0)	6/17 (37.5)	
HP 3–6 months after OP ^3^, n (%)			1
Improvement	1/17 (5.9)	1/17 (5.9)	
Steady state	6/17 (35.3)	6/17 (35.3)	
Deterioration	10/17 (58.8)	10/17 (58.8)	

HP = hypopituitarism, ICG = indocyanine green dye, OP = operation. ^1^
*p*-values for comparison of proportions between patient groups with ICG and without ICG ^2^ comparison of the three control measurements. ^3^ Comparison of preoperative and 3–6-month postoperative status.

**Table 4 diagnostics-14-01863-t004:** Number of affected axes before operation (OP) and 3–6 months after OP in the groups with and without ICG dye.

Number of Axes Affected	Before OP			3–6 Months after OP		
	Total	ICG	No ICG	Total	ICG	No ICG
0	14 (41.2)	8 (47.1)	6 (35.3)	8 (23.5)	5 (29.4)	3 (17.6)
1	5 (14.7)	2 (11.8)	3 (17.6)	6 (17.6)	3 (17.6)	3 (17.6)
2	10 (29.4)	3 (17.6)	7 (41.2)	7 (20.6)	0 (0)	7 (41.2)
3	3 (8.8)	3 (17.6)	0 (0)	10 (29.4)	8 (47.1)	2 (11.8)
4	2 (5.9)	1 (5.9)	1 (5.9)	3 (8.8)	1 (5.9)	2 (11.8)
*p*-value		0.3			0.013	

ICG = indocyanine green dye, OP = operation. Values in the table are frequencies (%). *p*-value for comparison between groups with and without ICG using Fisher–Freeman–Halton exact test.

**Table 5 diagnostics-14-01863-t005:** Comparison of the mean value of the number of affected axes between the group with ICG and the group without ICG.

Number of Axes Affected	Before OP				3–6 Months after OP				*p* ^2^
	Total	ICG	No ICG	*p* ^1^	Total	ICG	No ICG	*p* ^1^	
Average ± SE	1.24 ± 0.22	1.24 ± 0.34	1.24 ± 0.28	1	1.82 ± 0.23	1.82 ± 0.36	1.82 ± 0.30	1	0.003
Median (first, third quartile)	1 (0, 2)	1 (0, 2.5)	1 (0, 2)	0.919	2 (0.75, 3)	3 (0, 3)	2 (1, 2.5)	0.892	0.003

^1^—*p* value for the comparison between the ICG group and the non-ICG group using *t*-test for independent samples and Mann–Whitney test. ^2^—*p* value for the comparison between the total number of affected axes before and 3–6 months after OP with the *t*-test for the dependent sample and the Wilcoxon signed rank test. The interaction between the main factors ‘control measurement’ (before and 3–6 months after OP) and ‘patient group’ (with and without ICG) in the repeated measures analysis of variance model was statistically non-significant (*p* = 1). ICG = indocyanine green dye, OP = operation, SE = standard error.

**Table 6 diagnostics-14-01863-t006:** Type of affected axis before OP and 3–6 months after OP in groups with and without ICG dye.

Axis Type	Before OP			3–6 Months after OP		
	Total	ICG	No ICG	Total	ICG	No ICG
Tyroid axis	15	8	7	19	9	10
Adrenal axis	10	5	5	18	10	8
Gonadal axis	14	7	7	18	9	9
Somatotropic axis	2	1	1	7	3	4
Prolactin axis	0	0	0	0	0	0

The values in the table are absolute frequencies. ICG = indocyanine green dye, OP = operation.

**Table 7 diagnostics-14-01863-t007:** The impact of demographic and clinical characteristics on surgical outcome, assessed using a logistic regression model ^1^.

Factor	Hypopituitarism		Diabetes Insipidus	
	OR (95 % CI)	*p*	OR (95 % CI)	*p*
Age	1.04 (0.96–1.12)	0.337	1.03 (0.94–1.13)	0.518
Gender	2.86 (0.53–16.67)	0.223	2.18 (0.20–23.79)	0.522
Tumor size	1.00 (0.99–1.00)	0.101	1.00 (1.00–1.00)	0.169
Knosp	4.15 (0.96–17.86)	0.056	0.59 (0.16–2.13)	0.418
Hardy	2.48 (0.90–6.76)	0.078	0.53 (0.18–1.52)	0.235

ICG = indocyanine green dye, CI = confidence ratio, OR = odds ratio. ^1^
*p*-values for each factor are adjusted for the group of patients with and without ICG.

## Data Availability

The raw data supporting the conclusions of this article will be made available by the authors upon request.

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
