# Peer review of "Hypopituitarism, Diabetes Insipidus, and Syndrome of Inappropriate Antidiuretic Hormone Secretion after Pituitary Macroadenoma Surgery with Indocyanine Green Dye"

_diagnostics, 2024, doi:10.3390/diagnostics14171863_

Round 1

Reviewer 1 Report

Comments and Suggestions for Authors

General:

It is not very clear whether the sample size was calculated for the study groups, since this issue of the adequacy of the groups comes up several times in the text.

In the text, it is necessary to use a period as a separator between integers and decimal numbers.

In graphs 1-3, it is recommended to draw  lines and p-value between the columns.

Line:

17 and 103 - abbreviations as fT3, fT4

35 - triotropin - > thyrotropin

39 - the ellipsis should be replaced with specific syndromes or a period should be put

42 - the value is lost = 1 cm (also a macroadenoma).

59 - missing period

66 - first mention of ICG - introduction of abbreviation

117 - it is necessary to indicate what doses were prescribed as "prophylactic dose of hydrocortisone"

117 - minirin should be replaced with desmopressin with the desirable indication of the form of the drug

182 - Nomber should be corrected to Number

183 - It is recommended to present nonparametric data as median (first quartile, third quartile), and not as mean with standard error

214 - vs - it is recommended to write without a period, since this is an established abbreviation

215 - Table -> table

291 - 294 - it is not clear from what number are those %.  

Comments on the Quality of English Language

In the text, it is necessary to use a period as a separator between integers and decimal numbers.

In graphs 1-3, it is recommended to draw  lines and p-value between the columns.

Line:

17 and 103 - abbreviations as fT3, fT4

35 - triotropin - > thyrotropin

39 - the ellipsis should be replaced with specific syndromes or a period should be put

42 - the value is lost = 1 cm (also a macroadenoma).

59 - missing period

66 - first mention of ICG - introduction of abbreviation

117 - it is necessary to indicate what doses were prescribed as "prophylactic dose of hydrocortisone"

117 - minirin should be replaced with desmopressin with the desirable indication of the form of the drug

182 - Nomber should be corrected to Number

183 - It is recommended to present nonparametric data as median (first quartile, third quartile), and not as mean with standard error

214 - vs - it is recommended to write without a period, since this is an established abbreviation

215 - Table -> table

291 - 294 - it is not clear from what number are those %.  

Author Response

Dear Sir/Madam!
Thank you for taking the time to review our article and suggest constructive changes or corrections. All suggested corrections have been added to the text and highlighted in yellow to make them easier to find. I have attached the updated version. However, there are a few things that are not clear to me:

In graphs 1-3, it is recommended to draw lines and p-value between the columns - Did you mean to connect the tops of the columns with a line or to insert a line between each pair of columns? 
183 - It is recommended to present nonparametric data as median (first quartile, third quartile), and not as mean with standard error - In this table we have used both ways. Did you want us to delete the mean with standard error?

Comment: It is not very clear whether the sample size was calculated for the study groups, since this issue of the adequacy of the groups comes up several times in the text

Reply: Sample size was not calculated. We commented on the small sample size in the text because we expected to be able to include more patients with non-functional macroadenomas for the duration of the study. 

Reviewer 2 Report

Comments and Suggestions for Authors

In this prospective study, the authors analyzed the occurrence of postoperative hypopituitarism in 34 patients with pituitary macroadenomas who had their tumors removed either by fluorescence image-guided surgery (FIGS) or classically in a non-guided manner for a control group. In the FIGS method, the authors used the infusion of indocyanine green (ICG) dye to increase the discrimination between tumoral/pituitary tissue during the endoscopic transsphenoidal removal of the tumor. The ICG dye, which emits in the near-IR region of the electromagnetic spectrum, has a long history of intraoperative use as a visual aid to help differentiate between the more highly vascularized normal tissue and the less vascularized adenoma tissue. In the current study, the authors report that a lower proportion of patients developed diabetes insipidus 3-6 months post-operatively in the FIGS group compared to the control group. Conversely, more patients developed complications related to the function of adenohypophysis in the FIGS group compared to the control in the same time interval. None of these outcomes reached statistical significance. While the main hypothesis of the study and the methodology employed by the authors are sound, the small patient sample included in the study precludes any clear conclusions regarding the benefits of the ICG FIGS technique in preventing of post-operative complications in patients with pituitary macroadenomas. I have a few questions and/or comments for the authors as follows:

1.    Based on their experience, is it possible that smaller tumors (i.e., microadenomas) rather than larger ones would benefit more from the ICG FIGS technique?  As the authors stated, the differentiation between adenoma and normal pituitary tissue is already quite reliable under white light. However, while I am assuming that this is mostly true for larger tumors, it might not necessarily be for smaller ones and therefore the FIGS technique might offer a distinctive advantage in the latter case. The authors state that in a few cases in the control group (i.e., patients operated under white light) they encountered a piece of tissue that could not be easily identified as a healthy pituitary tissue or a tumor. How frequent were these occurences in the control group?  

2.    What is the residence time of the ICG dye in the circulation? According to the literature, the ICG dye binds tightly to plasma proteins and therefore becomes trapped in the vascular system. The authors state that the pituitary gland starts to fluoresce after about half a minute after the intravenous infusion of the dye, while the tumor tissue remains unstained. How long is this contrast maintained considering the differences in vascular permeability between the normal tissue and the tumor vasculature? 

3.  Did the authors look at the tumor recurrence rate for the patients enrolled in this study? Based on literature reports and their own experience, do the authors know if the ICG FIGS technique has any long-term impact on the tumor recurrence rate in patients with pituitary macroadenomas? Perhaps this could be added to the discussion section. 

4. Minor comment: there are several typos throughout the manuscript. For instance, in line 35 the words ‘corticotropin’ and ‘thyrotropin’ are misspelled. 

Author Response

Dear Sir/Madam!
Thank you for taking the time to review our article! Bellow are the answers to each one of your comments:

  1.  Based on their experience, is it possible that smaller tumors (i.e., microadenomas) rather than larger ones would benefit more from the ICG FIGS technique?  As the authors stated, the differentiation between adenoma and normal pituitary tissue is already quite reliable under white light. However, while I am assuming that this is mostly true for larger tumors, it might not necessarily be for smaller ones and therefore the FIGS technique might offer a distinctive advantage in the latter case. The authors state that in a few cases in the control group (i.e., patients operated under white light) they encountered a piece of tissue that could not be easily identified as a healthy pituitary tissue or a tumor. How frequent were these occurences in the control group?  Thank you for your comment! It is indeed possible that this technique would be more useful for functional microadenomas. In those cases, it is sometimes difficult to localize the tumour with certainty. However, such tumours are rarer and it would take much longer to collect a suitable sample. In principle, non-functional microadenomas do not cause any problems, so they are only followed up. As for doubtful cases, there was 1 such case in the control group and maybe 2 in the ICG group. Unfortunately, we did not make notes, which patient was it or record the surgery for future reference.

  1.  What is the residence time of the ICG dye in the circulation? According to the literature, the ICG dye binds tightly to plasma proteins and therefore becomes trapped in the vascular system. The authors state that the pituitary gland starts to fluoresce after about half a minute after the intravenous infusion of the dye, while the tumor tissue remains unstained. How long is this contrast maintained considering the differences in vascular permeability between the normal tissue and the tumor vasculature? - According to the literature, the half-life of ICG is 150 seconds. During surgery, the difference between the two tissues could be seen reliably for at least a few minutes. It would be difficult to say exactly how long, because we did not pay attention to this in our study. In principle, what was important for us was the first minute or two after the application, so that we could make a detailed inspection of the surgical area, define the border between the pituitary gland and the tumour, and then remove the tumour under white light. 

  1. Did the authors look at the tumor recurrence rate for the patients enrolled in this study? Based on literature reports and their own experience, do the authors know if the ICG FIGS technique has any long-term impact on the tumor recurrence rate in patients with pituitary macroadenomas? Perhaps this could be added to the discussion section. - We kept an eye out for residual or recurrent tumors with follow-up MR imaging 6 months after surgery and every 12 months thereafter. Patients who were included in the study at the beginning had already had some follow-up scans, while patients who were included in the study at the end had only had 1 follow-up scan so far. In this article we have focused more on the endocrinological aspects of ICG, and we plan to do an additional article in the future where we would focus more on the surgical aspects (intraoperative, postoperative complications in terms of CSF leak, infections, residual tumor or recurrence). To the best of our knowledge, there is no study yet reporting the long term impact on the tumour recurrence with the ICG FIGS technique. 

  1. Minor comment: there are several typos throughout the manuscript. For instance, in line 35 the words ‘corticotropin’ and ‘thyrotropin’ are misspelled. - Thank you for pointing this out. We have re-read the article and corrected any spelling mistakes we could find. I am reattaching the latest version of the article, where we have highlighted the corrections in yellow.

Round 2

Reviewer 2 Report

Comments and Suggestions for Authors

I thank the authors for their prompt responses to my comments. As I wrote in my previous report, while I find the study well-written and quite informative, it does not bring anything new to the field. A more successful patient accrual may or may not change the outcome but will certainly increase the power of the study. I am still somewhat hesitant to fully recommend the manuscript for publication in its present form. Nonetheless, I do recognize the utility of the work presented in this manuscript which could benefit potential readers. For this reason, I am changing my original recommendation regarding this manuscript. However, I defer the final decision to the editor.